# HARNESSING ORTHOGONALITY TO TRAIN LOW-RANK NEURAL NETWORKS

## ABSTRACT

In the realm of neural network training, the question of what is truly being learned beyond mathematical optimization has intrigued researchers for decades. This study delves into the essence of neural network weights. By leveraging the principles of singular value decomposition, we explore the hypothesis that the orthogonal bases of the low-rank decomposition of neural network weights stabilize during training, and provide experimental evidence to support this notion. Building upon this insight, we introduce Orthogonality-Informed Adaptive Low-Rank neural network training. Our novel approach seamlessly integrates into existing training workflows with minimal accuracy loss, as demonstrated by benchmarking on various datasets and well-established network architectures. We find that, through standard tuning procedures, our method surpasses the performance of conventional training setups. Finally, we showcase the effectiveness of our tuned low-rank training procedure by applying it to a state-of-the-art transformer model for time series prediction.

## 1 INTRODUCTION

When you train a neural network, what is it learning? From a technical view, its weights are iteratively adjusted based on the loss function's back-propagated gradients so as to minimize the difference between predicted outputs and target values. But what does this process look like from an intuitive standpoint? Does the mathematical optimization of the loss function impose a structure on the weights? Explaining why a neural network has learned something has intrigued researchers for decades.

A neural network's weights are typically represented as numerical values in a tensor. However, the sizes of these network weights has been increasing for a number of years Bernstein et al. (2021). One method to cope with these large networks on resource-constrained hardware is by representing them with low-rank decompositions Hsu et al. (2022). This approximation factorizes a full-rank matrix, $M$, into two or more matrices, where the inner dimension, $r$, is smaller than the dimensions of the original matrix. It is expressed as $M_{m \times n} = A_{m \times r} B_{r \times n}$, with the constraint that $r < \min(m, n)$. A common method for finding a low-rank approximation of a given matrix is singular value decomposition (SVD). SVD factorizes a matrix into an orthogonal basis $U$, an orthogonal cobasis $V$, and a diagonal matrix of the singular values sorted in descending order $\Sigma$, as $M = U\Sigma V^T$.

It has been shown multiple times that a neural network's weights can be effectively approximated by discarding the least significant singular values in their SVD, reducing the inner dimension of the decomposition Psichogios & Ungar (1994); Fontenla-Romero et al. (2018); Xue et al. (2014); Waleffe & Rekatsinas (2020). Many of these methods train on $U$, $V$, and $\Sigma$ but do not maintain the orthogonality of $U$ and $V$; those that do require additional training time Povey et al. (2018); Schotthöfer et al. (2022). Furthermore, low-rank methods often come at the cost of accuracy Ren & Xia (2023).

We posit that the orthogonal bases of the low-rank decomposition of neural network weights stabilize during training, while subsequent training focuses on the linear mixing of said basis. In this paper, we show experimental evidence of this hypothesis. We then demonstrate how we harness the beneficial attributes of orthogonality by carefully structuring the training process and thus ultimately producing more effective, streamlined models compared to the original full-rank models.

We provide implementations of the most common layer types and a method to wrap arbitrary model architectures for any learning task. We use this approach to train low-rank versions of common (state-of-the-art) networks. We find that, at best, our low-rank networks outperform their full-rank counterparts, and at worst show minimal accuracy loss.

Our contributions include:

- We show evidence that the multi-dimensional weights of a neural network stabilize during training.

- We propose a novel method of Orthogonality-Informed Adaptive Low-Rank (OIALR) neural network training and provide the tools for any researcher to use it on most networks.

- We demonstrate that our training approach seamlessly integrates into existing training workflows with minimal accuracy loss by means of benchmarks on multiple datasets, data modalities, well-known network architectures, and training tasks.

- We show that with tuning, OIALR outperforms conventional full-rank training.

## 2 RELATED WORK

The objective of network compression is to reduce a network's size and complexity while retaining or improving its performance Xu & McAuley (2023). For example, convolution layers have redundancies which can be removed Tai et al. (2016); Hssayni et al. (2022); Boufssasse et al. (2023). The most common methods of network compression use low-rank approximations. In its simplest form, a low-rank approximation factorizes a matrix $M$ as $M_{m \times n} = A_{m \times r} B_{r \times n}$ where $r < \min(m, n)$. This approach is applied in Cahyawijaya et al. (2021) where network weights are decomposed, then both $A$ and $B$ are trained with standard backpropagation.

The most common decomposition method for neural networks is SVD, see Wimmer et al. (2023); Cohen & Welling (2016); Nesky & Stout (2020); Psichogios & Ungar (1994); Guo et al. (2023), which comes with pre-existing optimized implementations and has been well-studied outside of the neural network landscape. To the best of our knowledge, the earliest usage of SVD in neural network compression is the SVD-NET Psichogios & Ungar (1994). This method reduces redundant hidden nodes within a simple network by decomposing the network weights with SVD and training $U$, $\Sigma$, and $V$. This basic approach has been utilized to great effect in many works, including addressing dataset imbalances Fontenla-Romero et al. (2018), last layer compression Sainath et al. (2013), entire network compression and training Xue et al. (2014), and sparse network training Swaminathan et al. (2020). While many of these methods do not maintain bases orthogonality, those that do show increased network performance Povey et al. (2018); Schotthöfer et al. (2022).

With the plethora of today's pre-trained models, researchers have spent substantial effort to fine-tune them for specific use cases, for which low-rank decompositions have been shown to be effective. In large language models (LLMs), LoRA Hu et al. (2022) demonstrated that adding a low-rank weight component alongside the originally trained weights can adapt LLMs to specialized use cases. DnA Jiang et al. (2022) takes a more extensive approach by reparameterizing the pre-trained model via a "self-supervised alignment step on the target domain using only unlabeled data before conducting the downstream supervised fine-tuning."

A version of the Conformer Gulati et al. (2020) was trained using a specialized low-rank decomposition using two matrices informed by the SVD of the original weight matrix Guo et al. (2023). This method, as well as Fontenla-Romero et al. (2018); Swaminathan et al. (2020); Cohen & Welling (2016); Ceruti et al. (2021); Hsu et al. (2022) and many of the approaches in Xu & McAuley (2023), actively train all low-rank matrices in the decomposition. In DLRT Schotthöfer et al. (2022), each of the three matrices in a model's singular-value decomposed weights is trained in a separate forward-backward pass while maintaining orthogonality. Beginning the training in low rank can be detrimental to network accuracy, as evidenced by previous work from Waleffe & Rekatsinas (2020); Bejani & Ghatee (2020). To correct for this, the corresponding methods opt to transition to a low-rank representation later in the training process. Furthermore, many low-rank methods show better generalization, i.e. reduced overfitting, during training Winata et al. (2020); Phan et al. (2020); Cahyawijaya et al. (2021).

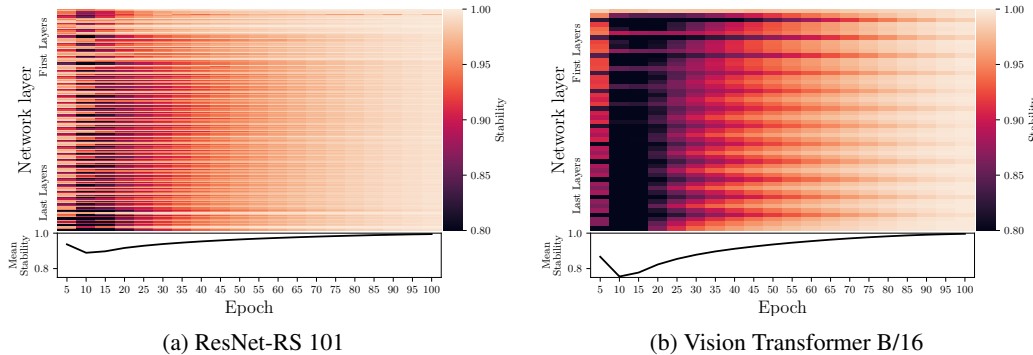

(a) ResNet-RS 101                             (b) Vision Transformer B/16

Figure 1: Stability, Equation (1), measurements of two different networks, both trained on ImageNet-2012 Russakovsky et al. (2015) measured at a frequency of five epochs, i.e., the bases of epoch $i$ is compared to the bases of epoch $i - 5$. X-axis is the epoch number, y-axis is the layer of the network with layers closest to the input at the top. Darker colors indicate less stability and lighter colors indicate more stability. Beneath each plot is the mean stability for each epoch shown.

When decomposing a matrix, its orthogonal basis can be viewed as the coordinate system in which the matrix operates. In this vein, it can be extremely useful for explaining the inner working of modern black-box neural networks. Intuitively, orthogonal networks are beneficial from an explainability viewpoint. ExNN Yang et al. (2021) utilized methods to preserve projection orthogonality during training to improve interpretability. Similarly, the Bort optimizer Zhang et al. (2023) aims to improve model explainability using boundedness and orthogonality constraints.

## 3    Observing Orthogonality Present in Neural Network Training

Like any real-valued, multi-dimensional matrix, each of a neural network's weights can be decomposed into an orthogonal matrix and a linear mixing matrix. We hypothesize that the orthogonal bases in the decomposition of the weights stabilize during training.

To test our hypothesis we determine each weight's orthogonal basis of different neural networks using the semi-orthogonal bases from its compact SVD. Compact SVD loosens the orthogonality constraints on $U$ and $V$ to only semi-orthogonal in the column dimension to reduce the inner dimension $r$. More precisely, it factorizes a matrix $M \in \mathbb{R}^{m \times n}$ into $U \Sigma V^T$, where $\Sigma \in \mathbb{R}^{r \times r}$ is a diagonal matrix with the singular values along its diagonal, $U \in \mathbb{R}^{m \times r}$ and $V \in \mathbb{R}^{n \times r}$ are semi-orthogonal matrices of orthogonal column vectors, and $r \leq \min(m, n)$. To obtain a two-dimensional (2D) representation of a weight tensor with more than two dimensions we maintain the original weight's leading dimension and collapse those remaining. If the resulting 2D weight representation is either not square or not tall and skinny, we use its transpose. We then find a compact SVD of this representation to obtain the orthogonal bases. While the SVD of $M$ is not unique, $U V^T$ is uniquely determined by $M (M^T M)^{-1/2}$ and is thus both unique and orthogonal, making it ideal for tracking.

To compare a weights's bases at two distinct times during training, $i$ and $j$, we define their relative stability $S_{ij}$ as the average of the dot products of each basis vector:

$$S_{ij} = \overline{(U V^T)_i \cdot (U V^T)_j} \tag{1}$$

where $(U V^T)_k$ represents the basis at a given time $k$ during training. Figure 1 shows how the layers' stability evolves for two vastly different network architectures, ResNet-RS 101 Bello et al. (2021) and the VisionTransformer (ViT) B/16 Dosovitskiy et al. (2021), both trained on ImageNet-2012 Russakovsky et al. (2015). In both cases, the bases stabilize during training, as shown by the increase in the relative stability $S$ over the training time.

There are general patterns which quickly emerge from these figures: the weights closer to the output stabilize faster than those toward the input, the stabilization rate differs depending on the layer

---

**Algorithm 1:** Updating a weight's basis and cobasis, $\boldsymbol{U}$ and $\boldsymbol{V}$, from the weight's actively trained singular value matrix $\boldsymbol{\Sigma}$. $\boldsymbol{U}$, $\boldsymbol{\Sigma}$, and $\boldsymbol{V}$ comprise the current low-rank SVD of with weight $\mathbf{W}$, variables with a $'$ are found from the trained values of $\boldsymbol{\Sigma}$.

---

**Input:** Basis $\boldsymbol{U}$, cobasis $\boldsymbol{V}$, trained $\boldsymbol{\Sigma}$ matrix

1 $\boldsymbol{U}', \boldsymbol{\Sigma}', \boldsymbol{V}'^{T} \leftarrow \text{SVD}(\boldsymbol{\Sigma})$        // $\mathbf{W} \approx \boldsymbol{U}\boldsymbol{\Sigma}\boldsymbol{V}^{T}$

2 $\boldsymbol{U} \leftarrow \boldsymbol{U}\boldsymbol{U}', \boldsymbol{V}^{T} \leftarrow \boldsymbol{V}'^{T}\boldsymbol{V}^{T}, \boldsymbol{\Sigma} \leftarrow \boldsymbol{\Sigma}'$        // $\mathbf{W} \approx \boldsymbol{U}\boldsymbol{U}'\boldsymbol{\Sigma}'\boldsymbol{V}'^{T}\boldsymbol{V}^{T}$

---

type, and the stability increases throughout training. These observations intuitively make sense: the updates for weights closer to the network's output will have fewer layer terms thus the weights will get more direct responses for their 'mistakes,' a linear layer behaves differently than a convolution or attention layer, and as we move towards the end of training each layer has learned most of how it responds to inputs.

We theorize that the stabilization of the orthogonal bases during network training is crucial to the optimization process. Intuitively, this process resembles language learning Chomsky (1972). First, you develop a feeling for the language by acquiring a basic vocabulary and some phrases. Then, you delve into more complex grammar. Once you have learned the grammar, there is no need to re-learn it from scratch, you only need to learn how to apply it. Similarly, neural networks first grasp the task's fundamentals, then discern the roles of each layer, and ultimately fine-tune each layer's functionality for its task.

## 4 ORTHOGONALITY-INFORMED ADAPTIVE LOW-RANK TRAINING

To harness the stabilized orthogonal bases in neural network training, we present a novel algorithm that reduces the number of trainable parameters while maintaining accuracy and overall time-to-train.

As shown in Figure 1, most layers' bases do not stabilize before a few epochs have passed. Therefore, we start training in a traditional full-rank scheme. After a specified number of iterations $d$, we transition the network's eligible weights to low rank via their SVD. Experimentally, we found that the delay should be about one third of the total number of iterations. At this point, we no longer train $\boldsymbol{U}$ and $\boldsymbol{V}^{T}$ with backpropagation and train only the square matrix $\boldsymbol{\Sigma}$. After a specified number training steps $\nu$, the bases $\boldsymbol{U}$ and $\boldsymbol{V}^{T}$ are updated by extracting the new bases from the trained $\boldsymbol{\Sigma}$ matrix using an SVD of $\boldsymbol{\Sigma}$, as outlined in Algorithm 1. After the basis $\boldsymbol{U}$ and cobasis $\boldsymbol{V}^{T}$ are updated, a new inner rank is found by removing all singular values whose absolute magnitude is less than $\beta$ times the largest singular value in the current $\boldsymbol{\Sigma}$, where $\beta$ is a hyperparameter that defaults to $0.1$. As the first layers of the network are unstable for longer and likely require most of their ranks, the update of $\boldsymbol{U}$ and $\boldsymbol{V}$ is only applied to the last $\ell$ layers of the network where $\ell = L \cdot \alpha \cdot u$, where $L$ is the number of network layers, $\alpha$ is a hyperparameter defaulting to $0.1$, and $u$ is the number of updates which have been completed. This process repeats until the end of training. Optionally, the first or last layers can be excluded from low-rank training depending on the use case. We provide an outline of our Orthogonality-Informed Adaptive Low-Rank (OIALR) training in Algorithm 2.

## 5 EXPERIMENTS

To validate our OIALR training approach, we conducted several experiments using different neural network architectures and datasets. We focused on demonstrating its effectiveness in terms of reducing the number of trainable parameters while maintaining, or improving, network performance and training time. We began by naively applying the OIALR method to a standard neural network training setup to explore what a typical researcher would experience in Sections 5.2 to 5.4. However, as OIALR changes the network structure during training, we expect the hyperparameters to vary from those commonly used for full-rank training. For the final two experiments, Sections 5.5 and 5.6, we tuned the hyperparameters for OIALR using `Propulate` Taubert et al. (2023), an asynchronous evolutionary optimization algorithm designed for usage on high performance clusters which has been shown to be effective for neural architecture searches Coquelin et al. (2021).

---

**Algorithm 2:** OIALR training method on an unspecified model $M$.

---

**Input:** Model $M$, training steps $t_{\max}$, delay $d$, low-rank update frequency $\nu$, singular value cutoff fraction $\beta$, percentage of layers in each low-rank update step size $\alpha$

1  $L \leftarrow$ Number of possible low-rank weights in $M$
2  $\ell \leftarrow L \cdot \alpha$
3  **for** $t \leftarrow 1$ **to** $t_{\max}$ **do**
4      **if** $t < d$ **then**
5          Train full-rank network.
6      **else if** $t = d$ **then**
7          Convert network to low rank.
8      **else if** $t \mod \nu = 0$ **then**
9          **for** $i \leftarrow L - \ell$ **to** $L$ **do**
10             Update weight $i$ with Algorithm 1.
11             Remove singular values $< \beta \cdot \texttt{max}(\mathbf{\Sigma}_i)$.
12             Reshape $\boldsymbol{U}_i$, $\boldsymbol{V}_i$, $\mathbf{\Sigma}_i$, and optimizer states.
13         $\ell \leftarrow \ell + L \cdot \alpha$
14     **else**
15         Train low-rank network ($\mathbf{\Sigma}$ for all low-rank weights).

---

Table 1: Baseline and OIALR trainings of ViT-B/16 and ResNet-RS 101 Bello et al. (2021) on ImageNet-2012 Russakovsky et al. (2015). 'Time' refers to the time to train in hours, the last row shows the number of trainable parameters in the low-rank model as a percentage of the full-rank model.

| | ViT-B/16 | | | | ResNet-RS 101 | | | |
|---|---|---|---|---|---|---|---|---|
| | Loss | Top-1 | Top-5 | Time | Loss | Top-1 | Top-5 | Time |
| Baseline | **2.16** | **71.64 %** | **89.18 %** | 3.29 h | **1.78** | **78.75 %** | **94.21 %** | **5.55 h** |
| OIALR | 2.20 | 70.30 % | 88.73 % | **3.26 h** | 1.81 | 77.95 % | 93.95 % | 5.92 h |
| Parameters | **16.56 %** | | | | **15.66 %** | | | |

In an attempt to showcase how our method would perform on real-world use cases, our experiments used state-of-the-art techniques and models, including strong image transforms Touvron et al. (2021), dropout Srivastava et al. (2014), learning rate warm-up Gotmare et al. (2019), and cosine learning rate decay Loshchilov & Hutter (2017), the utilized implementations of which are from Wightman et al. (2023). We trained all networks using the AdamW Loshchilov & Hutter (2018) optimizer. The complete sets of hyperparameters are included in Appendix A. We list results as the average of three runs, all of which have unique random seeds.

## 5.1  COMPUTATIONAL ENVIRONMENT

We ran all experiments on a distributed-memory, parallel hybrid supercomputer. Each compute node is equipped with two 38-core Intel Xeon Platinum 8368 processors at $2.4\,\text{GHz}$ base and $3.4\,\text{GHz}$ maximum turbo frequency, $512\,\text{GB}$ local memory, a local $960\,\text{GB}$ NVMe SSD disk, two network adapters, and four NVIDIA A100-40 GPUs with $40\,\text{GB}$ memory connected via NVLink. Inter-node communication uses a low-latency, non-blocking NVIDIA Mellanox InfiniBand 4X HDR interconnect with $200\,\text{Gbit/s}$ per port. All experiments used Python 3.10.6 with `CUDA`-enabled `PyTorch` 2.0.0 Paszke et al. (2019). The source code for the implementation is publicly available[1].

## 5.2  VISION TRANSFORMER ON IMAGENET-2012

For the first experiment, we trained the Vision Transformer (ViT)-B/16 model Dosovitskiy et al. (2021) in its standard form on the ImageNet-2012 dataset Beyer et al. (2020) using the ReaL validation labels Beyer et al. (2020). The considerable number of parameters in this model provided

---

[1]Will be released upon publication

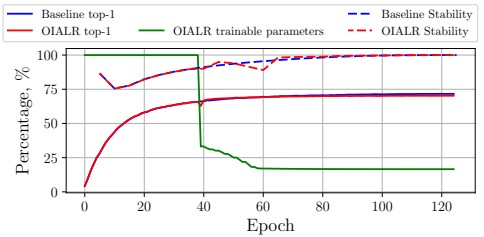 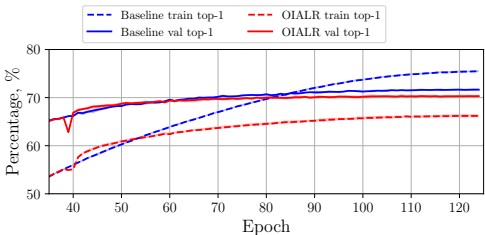

(a) Top-1 validation accuracy, percentage of trainable parameters as opposed to the full-rank network, and average stability of the bases.

(b) Top-1 accuracies for training and validation with baseline and OIALR training methods.

Figure 2: Training curves from training a ViT-B/16 network on ImageNet-2012 for 125 epochs.

us with a rigorous testing ground for our OIALR training method. We maintained the same hyper-parameters for both full-rank and adaptive low-rank. To reduce the environmental impact of our experiments, we trained for 125 instead of the original 300 epochs Dosovitskiy et al. (2021). At this point in training, the validation accuracy was observed to be nearly stabilized, see Figure 2b. In this vein, we used an image resolution of $160 \times 160$ instead of $224 \times 224$ to reduce the training time and energy consumption.

The results are shown in Figure 2 and Table 1. Figure 2a illustrates the top-1 validation score, the number of trainable parameters as a percentage of the full-rank model, and the average network stability (as in Figure 1) throughout the training. We observe that, while the baseline stability increases smoothly throughout the training, OIALR's stability is less consistent as the ranks of the weights are reduced. This is caused by the reduction of ranks themselves as the $UV^T$ from five epochs prior contains more basis vectors than the current $UV^T$. An important observation is the momentary drop in accuracy when the network switches from full-rank to low-rank weights, best seen in Figure 2b, although it quickly rebounds to higher accuracies than before. We theorize that this is caused by the left-over momentum states in the optimizer.

OIALR training reduced the training time by approximately $1\%$, though the resulting low-rank model showed $1.34 \pm 0.39\%$ worse performance in this naive configuration. The OIALR method decreased the number of trainable parameters to $16.56 \pm 0.23\%$ of the full-rank parameters. Figure 2b shows that the full-rank model has moved into the overfitting regime, where the training accuracy continues increasing while the validation accuracy does not, and the low-rank model has not.

## 5.3 RESNET-RS 101 ON IMAGENET-2012

To show the versatility of our approach, we trained the ResNet-RS 101 Bello et al. (2021) architecture on ImageNet-2012 for image classification. We trained the model for 125 epochs with an image resolution of $160 \times 160$ and validate using a resolution of $224 \times 224$ on the ReaL labels Beyer et al. (2020). The results are included in Table 1.

At the end of OIALR training, only $15.66 \pm 0.08\%$ of the original full-rank weights remain trainable with only a minor $1.03 \pm 0.16\%$ reduction in top-1 accuracy. Due to implementation limitations and the conflicting 2D weight representation and 2D convolution operation, low-rank training was $6.67 \pm 0.49\%$ slower than full-rank training.

## 5.4 ONEFORMER ON CITYSCAPES

In the next experiment, we train the OneFormer Jain et al. (2023) model on the CityScapes Cordts et al. (2016) dataset for semantic segmentation. OneFormer is "a universal image segmentation framework that unifies segmentation with a multi-task train-once design" Jain et al. (2023), i.e. this model trains on multiple tasks at once. We trained the model with the configuration provided by the original authors with only slight modifications (reduced batch size to fit into GPU VRAM and model wrapping for OIALR runs). The results are shown in Table 2.

Table 2: Training the OneFormer on the CityScapes dataset. The rightmost column shows the number of trainable parameters as a percentage of the full-rank model.

|  | IoU | Category IoU | Time to train | Trainable Parameters |
|---|---|---|---|---|
| Full rank | **72.47 %** | **88.68 %** | **32.76 h** | 100 % |
| OIALR | 68.18 % | 87.32 % | 32.95 h | **28.56 %** |

Table 3: A mini ViT similar to that used in Hassani et al. (2022) trained on CIFAR-10. Baseline indicates the full-rank trainings. 'OAILR, tuned' trainings used tuned hyperparameters, while 'OIALR, untuned' used the same hyperparameters as the baseline. Accuracies and loss values are determined on the test dataset. The rightmost column shows the number of trainable parameters as a percentage of the full-rank model.

|  | Loss | Top-1 | Top-5 | Time to train | Trainable Parameters |
|---|---|---|---|---|---|
| Full rank | 0.88 | 85.17 % | 98.34 % | 12.14 min | 100 % |
| OIALR, untuned | 0.91 | 83.05 % | 98.38 % | 11.99 min | 30.98 % |
| OIALR, tuned | **0.85** | **86.33 %** | **98.53 %** | **11.19 min** | **9.97 %** |

As expected, when using hyperparameters designed for full-rank training, we note a reduction in the class IoU of $4.29 \pm 0.21\,\%$ and a slight reduction in the categorical IoU of $1.02 \pm 0.09\,\%$. While the OIALR model had $28.56 \pm 0.02\,\%$ of the full-rank model's trainable parameters, it required $0.58 \pm 0.02\,\%$ more time to train, this equates to $11.5\,\mathrm{min}$ longer in wallclock time.

## 5.5 ABLATION STUDY ON MINI ViT ON CIFAR-10

To show how OIALR performs with proper tuning, we trained a reduced-size ViT model on the CIFAR-10 dataset with and without tuning. The runs without tuning utilize the same hyperparameters as the baseline runs. As reduced size ViT models have been shown to perform superbly Hassani et al. (2022) at a fraction of the compute time, we elect to use a ViT-B/16 variant with a patch size of 8, 6 layers, and 6 attention heads in this experiment (original values are a patch size of 16, 12 layers, and 12 attention heads). The results of this experiment are shown in Table 3 and Figure 3.

The top-1 test results are shown in Figure 3a alongside the stability measurements and number of trainable parameters for both the tuned OAILR runs and the baseline runs. We can see here that the baseline stability is nearly constant the entire training, indicating that the network was only tuning the linear mixing of the bases after the fifth epoch. The stability of OIALR runs shows a saw-tooth pattern each time the rank was lowered as the previous bases was composed with basis vectors which had since been removed.

Interestingly, the best learning rate schedule for the OIALR trainings which was found in the hyperparameter search increases the learning rate as the number of parameters is reduced, see Figure 3b.

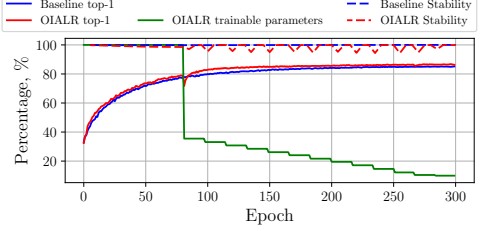 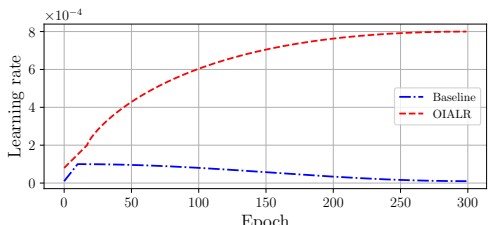

(a) Top-1 test accuracy, trainable parameters as the percentage of the full-rank model, and average stability of the bases with a five epoch frequency.

(b) Learning rate schedules for baseline training and OIALR training. OIALR training learning rate schedule determined by hyperparameter search.

Figure 3: Training curves for a mini ViT on CIFAR-10 for tuned OIALR and baseline trainings.

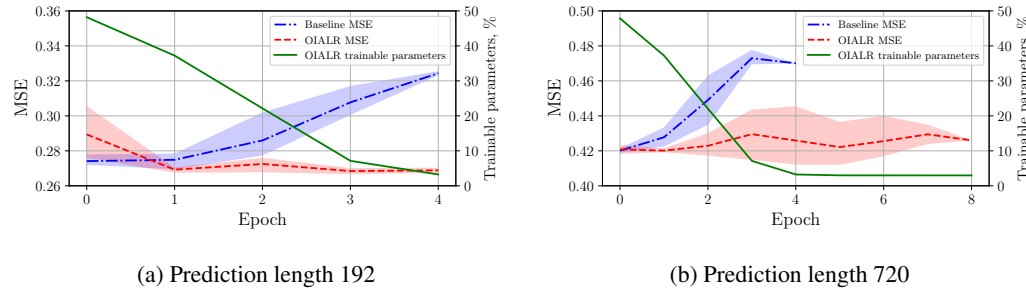

| (a) Prediction length 192 | (b) Prediction length 720 |

Figure 4: MSE and percentage of trainable parameters as opposed to the full-rank network for the Autoformer trained on the ETTm2 dataset using two different prediction lengths in $15\,\mathrm{min}$ time steps.

This result makes intuitive sense: as the number of trainable parameters decreases, the learning rate applied to the gradients of the remaining parameters can be increased without the model degrading.

The tuned OIALR model reduced the number of trainable parameters by $90.03 \pm 0.13\,\%$ while increasing predictive performance over the baseline from $85.17 \pm 0.42\,\%$ to $86.33 \pm 0.71\,\%$. Furthermore, training time was reduced by $8.52 \pm 0.82\,\%$ over the baseline. Although the untuned OIALR model reduced the number of trainable parameters by $69.02 \pm 0.07\,\%$, the top-1 test accuracy drops by over $2\,\%$.

## 5.6 ABLATTION STUDY ON AUTOFORMER ON ETTM2

This use case serves as a crucial test for OIALR training as it demonstrates the method's versatility by applying it to a model in a radically different domain. Furthermore, it serves to validate that the findings depicted in Figure 1 remain applicable in non-image scenarios.

The Electricity Transformer Dataset Zhou et al. (2021) (ETT) measures load and oil temperature of electrical transformers. It contains 70,000 measurements, available in different levels of granularity, each with seven different features and is primarily used for time series forecasting. We focus on the ETTm2 dataset, which uses a 15-minute resolution. Common prediction lengths for this dataset are 96, 192, 336, and 720 time steps. The Autoformer Wu et al. (2021) is a well-known transformer model contained in Hugging Face's repository. Unlike the other tested transformers, this model uses auto-correlation layers and one-dimensional convolutions. Due to its success at the time, it was deployed at the 2022 Winter Olympics for weather forecasting.

As the baseline for this experiment suffers quickly from overfitting, see Figure 4, we start in low rank instead of transitioning during training. Although we do see some overfitting in the OIALR results, it is much less severe than in the baseline. As Table 4 indicates, the tuned OIALR models

Table 4: Training of the Autoformer model on the ETTm2 dataset. Baseline and untuned OIALR hyperparameters were chosen as the default parameters in the original source. Tuned OIALR hyperparamters found via `Propulate`. Prediction lengths are in $15\,\mathrm{min}$ time steps. The optimal values for the mean squared error (MSE) and mean absolute error (MAE) are both zero. The last two rows show the number of trainable parameters as a percentage of the full-rank model for untuned and tuned OIALR training respectively.

| Prediction length | 96 | | 192 | | 336 | | 720 | |
|---|---|---|---|---|---|---|---|---|
| | MSE | MAE | MSE | MAE | MSE | MAE | MSE | MAE |
| Base | 0.2145 | 0.2994 | 0.2737 | 0.3356 | 0.3277 | 0.3640 | 0.4194 | 0.4157 |
| OIALR, untuned | 0.2140 | 0.2974 | 0.2773 | 0.3336 | 0.3253 | 0.3632 | 0.4213 | 0.4186 |
| OIALR, tuned | **0.2112** | **0.2942** | **0.2686** | **0.3305** | **0.3212** | **0.3591** | **0.4120** | **0.4147** |
| Parameters, untuned | 49.09 % | | 43.59 % | | 45.67 % | | 51.33 % | |
| Parameters, tuned | **27.53 %** | | **8.85 %** | | **4.88 %** | | **4.47 %** | |

were more accurate for all prediction lengths with a drastically decreased number of parameters, The untuned OIALR models outperformed the baseline in some cases, and succeeded in reducing the number of trainable parameters by $47.42\,\%$ on average. Interestingly, the tuned OIALR model required more trainable parameters for predicting shorter time-spans.

In contrast to the previous experiment, the best learning rate scheduler found for this use case more closely resembles a traditional scheduler, featuring a warm-up phase followed by a gradual decay. This may be related to the fact that the networks, both low-rank and full-rank, overfit the training dataset quickly.

## 6 CONCLUSION

There has long been curiosity about what is being learned within a neural network during training. This study aimed to shed light on this question by exploring the nature of neural network weights during training through their singular value decomposition. Our findings revealed that the orthogonal component of the a neural network's weights stabilize early in the training process. Building on this discovery, we introduced Orthogonality-Informed Adaptive Low-Rank (OIALR) training.

We used the OIALR training method to train low-rank versions of common and state-of-the-art networks across various data modalities and tasks. While our method may not improve upon traditional training techniques per se, it can outperform them in terms of accuracy and time to train when tuned appropriately. Notably, its true potential lies in significantly reducing the number of trainable parameters, enabling model fine-tuning and production on resource-constrained devices and reducing the amount of data to communicate during distributed training.

Integrating orthogonality-based training methods into the deep learning researcher's toolkit offers promising possibilities for a wide range of applications. With this work, we hope to inspire further exploration and refinement of orthogonality-informed methods, ultimately advancing the field of machine learning and its practicality across diverse domains.

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
