# Harnessing Orthogonality to Train Low-Rank Neural Networks: Supplementary Material

## 1 Experiment Hyperparameters

Parameters not listed use the default values in the respective implementations.

### 1.1 ImageNet-2012

The non-default hyperparameters for all experiments on the ImageNet-2012 dataset are shown in Table 1. We utilized the ViT implementation from Torchvision Paszke et al. (2019) and the ResNet-RS 101 implementation from Wightman et al. (2023).

### 1.2 OneFormer

As the hyperparameter for OneFormer are pulled directly from the original source Jain et al. (2023), we list only the place where we differ. We utilized `Base-Cityscapes-UnifiedSegmentation.yaml` as our base configuration. The other hyperparameters are listed in Table 2.

### 1.3 Mini-ViT on CIFAR-10

For training the mini-ViT we used most of the same parameters as listed in Table 1 except a lower learning rate. The utilized ViT for these experiments was from Wightman et al. (2023). The training parameters for these experiments are shown in Table 3. The search space for `Propulate` and the parameters for the search itself are shown in Table 4

### 1.4 AutoFormer on ETTm2

The learning rate schedule used in the original AutoFormer Wu et al. (2021) is a step-based schedule with fixed steps, it is denoted as 'type1.' The hyperparameters used in our experiments are listed in Table 5. The parameters for the hyperparameter search are listed in Table 6.

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

Table 1: Hyperparameters for training networks on ImageNet-2012. Dataset parameters are referring to the dataset transforms provided by Wightman et al. (2023). LR k-decay is a parameter of the cosine learning rate decay Zhang & Li (2022)

| General Training Hyperparameters | | | |
|---|---|---|---|
| Local batch size | 128 | Learning Rate Scheduler | |
| Global batch size | 1024 | Learning rate (LR) | 0.001 |
| Autocast to bfloat16 | True | Minimum learning rate | 0.00001 |
| Epochs | 125 | Warmup LR | 0.00001 |
| Label smoothing | 0.1 | LR k-decay | 1 |
| Optimizer | AdamW | Warmup epochs | 10 |
| Sync batchnorm | True | | |
| General dataset hyperparameters | | | |
| Interpolation | random | Auto augment | rand-m15-mstd0.5-inc1 |
| Random erasing probability | 0.25 | crop pct | 0.9 |
| Random erasing mode | pixel | scale | (0.08, 1) |
| | | Training crop size | 160 |
| OIALR hyperparameters | | | |
| Full rank first layer | False | Delay | 25000 |
| Stability frequency | 1000 | Full rank last layer | True |
| Sigma cutoff fraction | 0.1 | | |
| ResNet-RS 101 hyperparameters | | | |
| Dropout | 0.25 | Validate crop size | 224 |
| ViT B/16 hyperparameters | | | |
| Dropout | 0.1 | Hidden dim | 768 |
| Mlp dim | 3072 | Num layers | 12 |
| Num heads | 12 | Patch size | 16 |

Table 2: Hyperparameters used for training OneFormer models on the CityScapes dataset for segmentation

| | |
|---|---|
| cfg.SOLVER.IMS_PER_BATCH | 12 |
| cfg.MODEL.SVD_STABILITY_FREQUENCY | 500 |
| cfg.MODEL.SVD_DELAY | 30000 |
| cfg.MODEL.SVD_UVHTHRESHOLD | 0.99 |
| cfg.MODEL.SVD_SIGMA_CUTOFF_FRACTION | 0.4 |
| cfg.MODEL.SVD_KEEP_FIRST_LAYER | True |
| cfg.MODEL.SVD_KEEP_LAST_LAYER | True |
| cfg.MODEL.TEST.TASK | semantic |

Table 3: Hyperparameters used for CIFAR-10 training runs. General hyperparameters used for all runs, OAILR hyperparamters use for all OAILR runs. Dataset parameters refer to implementation options in `timm` Wightman et al. (2023)

| General Hyperparamters | | | |
|---|---|---|---|
| Train crop size | 32 | Label smoothing | 0.1 |
| Local batch size | 256 | Optimizer | AdamW |
| Global batch size | 1024 | auto_augment | rand-m9-mstd0.5-inc1 |
| Autocast to bfloat16 | True | Crop percent | 1 |
| Random erasing probability | 0.25 | Image scale | (0.8, 1.0) |
| Random erasing mode | pixel | Interpolation | random |
| ViT depth | 6 | ViT num heads | 6 |
| ViT qkv_bias | False | ViT patch_size | 8 |
| ViT embed_dim | 768 | ViT drop_path_rate | 0.2 |
| ViT mlp_ratio | 4 | | |
| Baseline | | Tuned | |
| LR | 0.0001 | LR | 0.0002 |
| Minimum LR | 0.00001 | Minimum LR | 0.0008 |
| Warmup LR | 0.00001 | Warmup LR | 0.00008 |
| LR k-decay | 1 | LR k-decay | 0.4 |
| Warmup epochs | 10 | Warmup epochs | 17 |
| OIALR hyperparameters | | | |
| Delay | 4000 | Stability frequency | 1000 |
| Full rank last layer | True | Sigma cutoff fraction | 0.2 |
| Full rank first layer | False | | |

Table 4: Propulate search parameters for the mini ViT on CIFAR-10

| Parameters to search over | Search space | Propulate parameter | Value |
|---|---|---|---|
| LR | (5e-5, 1e-3) | Crossover probability | 0.7 |
| Minimum LR | (5e-6, 1e-3) | Mutation probability | 0.4 |
| Warmup LR | (5e-6, 2e-4) | Random init probability | 0.1 |
| LR k-decay | (0.1, 2) | Number of islands | 8 |
| Warmup epochs | (1, 20) | Migration probabilty | 0.9 |
| OIALR sigma cutoff fraction | (0.01, 0.9) | | |
| OIALR stability frequency | (200, 1000) | | |
| OIALR delay | (10e2 10e3) | | |

Table 5: Hyperparameters used for training AutoFormer models on the ETTm2 dataset.

| General Hyperparameters | | | |
|---|---|---|---|
| Dimension of linear layers | 2048 | Number encoder layers | 2 |
| Loss function | MSE | Early stopping patience | 3 |
| Decoder input size | 7 | Start token length | 48 |
| Use distilling | True | Activation function | gelu |
| Encoder input size | 7 | Batch size | 32 |
| Attention factor | 1 | Moving average window | 25 |
| Dimension of model | 512 | Maximum training epochs | 20 |
| Dropout | 0.05 | Output attention | False |
| Number of heads | 8 | Number decoder layers | 1 |
| Default LR schedule | | Tuned LR schedule | |
| LR schedule | type1 | LR schedule | cosine |
| Learning rate | 0.0004 | learning_rate | 0.01 |
| | | lr_k_decay | 0.85 |
| | | min_lr | 0.0004 |
| | | warmup_lr | 0.0001 |
| | | warmup_epochs | 3 |
| OIALR Hyperparameters | | | |
| Delay | 600 | Full rank first layer | True |
| Full rank last layer | True | Stability frequency | 400 |
| Full rank warmup | False | Sigma cutoff fraction | 0.4 |

Table 6: The search space and settings for the hyperparameter search using `Propulate`.

| Parameter | Search Space | Propulate Parameter | Value |
|---|---|---|---|
| LR | (1e-5, 5e-3) | Crossover probability | 0.7 |
| Minimum LR | (5e-6, 1e-3) | Mutation probability | 0.4 |
| Warmup LR | (5e-6, 2e-4) | Random init probability | 0.1 |
| LR k-decay | (1e-3, 2) | Number of islands | 4 |
| Warmup epochs | (2, 10) | Migration probabilty | 0.9 |
| OIALR sigma cutoff fraction | (0.01, 0.9) | | |
| OIALR stability frequency | (50, 2000) | | |
| OIALR delay | (250, 2500) | | |
| OIALR full rank first layer | (False, True) | | |
| OIALR full rank last layer | (False, True) | | |