# OpenReview forum: "Harnessing Orthogonality to Train Low-Rank Neural Networks"
_ICLR.cc/2024/Conference — Submitted to ICLR 2024_

### Official Review · Reviewer_Mh5x · 2023-10-30

**Soundness:** 3 good
**Presentation:** 3 good
**Contribution:** 2 fair
**Rating:** 5
**Confidence:** 4

**Summary:**

The paper starts by defining a notion called relative stability and illustrating the stability of network weights during training tends to plateau early on. Such observation motivates the authors to develop the method Orthogonality-Informed Adaptive Low-Rank (OIALR). This approach starts with an SVD of the weight matrices ($U \Sigma V^T$) and only updates $\Sigma$ at the majority time of training. This strategy significantly trims the number of training parameters. Empirical data further substantiates that the OIALR method either matches or marginally underperforms when compared with the baseline full-rank training, yet is less prune to overfitting.

**Strengths:**

1. The paper is overall well-written and clearly motivated.
2. The proposed method which only updates the $\Sigma$ matrix is quite interesting.
2. The reviewer personally appreciates the authors for reporting the untuned results.

**Weaknesses:**

A major contribution listed in the paper is reducing the number of trainable parameters during training hence allowing shorter training time and enabling fine-tuning and production on resource-constrained devices (as stated by the authors in the Conclusion section). Yet, this assertion seems not well substantiated by the experiments: (1) based on the experiments listed in the paper, OIALR shows nearly identical training time as the baseline and sometimes requires even longer time; (2) concerning memory allocation, the OIALR method, especially at step 7 in Algorithm 2, demand more memory than the baseline (factorizing a matrix to 3 matrices increase its memory cost) and throughout the training for a long time until $r$ is decreased to some small value. Hence the resource reduction is questionable; (3) if we instead consider the RAM consumption during training (given OIALR's reduced parameter count), there might indeed be potential savings. However, the paper lacks empirical validation in this context (e.g., a table to compare the RAM used between OIALR and the baseline). Given these observations, the reviewer finds it hard to evaluate the contribution of the paper as of the current version.

**Questions:**

1. In Figure 1, the stability measure exhibits an initial decline followed by a subsequent rise. What could account for this initial drop in the early stages?
2. In some of the experiments shown in the paper, the authors show the results for both tuned and untuned OIALR, but not in other experiments, which is a bit confusing to the reviewer. Is this due to time or resource constraints?
3. To clarify, in Section 3, the authors wrote "To obtain a two dimensional (2D) representation of a weight tensor with more than two dimensions we maintain the original weight’s leading dimension and collapse those remaining", does this mean that for a tensor with dimension $a \times b \times c$, it will be transformed to a matrix with dimension $ a \times (bc) $?

---

> ### Author Response · Authors · 2023-11-20
>
> - The statement on enabling fine-tuning and production on resource-constrained devices refers to the trained model which uses the stable bases found during training. If a use-case desires to use a pre-trained model, they do not need to retrain most of the basis, but only tune it. Similar to what is happening later on in training.
> - Regarding RAM usage, the implementation of the model is not perfect at this point. Theoretically, an optimized version of OIALR training would requires less memory during both the forward and backward passes so long as the inner dimension of the layer is small. Furthermore, the states of the optimizer would not need to be stored for all layers. The resulting savings can be quite large in the end, given an optimized implementation.
> Q1: The initial drop in Stability show in Figure 1 is attributed to the network 'learning' that is needs to adjust the basis. At first, it attempts to fix itself locally, using the random bases provided at initialization. When this is unsuccessful, it then learns to change the basis.
> Q2: Unfortunately, to tune the other experiments would be too resource and time intensive. To turn the small experiments, it took the hyperparameter optimizer hundreds of training runs. To do this for multiple methods which take multiple hours and are more complex, would takes much too much compute and wall-clock time.
> Q3: You are correct.

---

### Official Review · Reviewer_LcGV · 2023-10-31

**Soundness:** 2 fair
**Presentation:** 3 good
**Contribution:** 2 fair
**Rating:** 3
**Confidence:** 4

**Summary:**

This paper investigates the hypothesis that the orthogonal bases of the low-rank decomposition of neural network weights stabilize during training. The authors introduce Orthogonality-Informed Adaptive Low-Rank (OIALR) neural network training, which seamlessly integrates into existing training workflows with minimal accuracy loss. Experimental evidence is provided to support the hypothesis, and the effectiveness of the OIALR training approach is demonstrated through benchmarking on various datasets and network architectures.

**Strengths:**

1. The paper is easy to follow.
2. The idea of orthogonal bases of the low-rank decomposition is reasonable.

**Weaknesses:**

1. The experimental results presented seem limited, with only two models tested and no comparisons to prior work. This makes it challenging to verify the effectiveness of the proposed method. It would be beneficial to include additional models and draw comparisons with previous works.
2. The algorithm initially trains the full-rank network during the initial epochs. As such, it might be more suitable for the term "full-rank to low-rank training" rather than strictly low-rank training. And what's the effect of removing the full-rank training phase?
3. How does this method compare in terms of advantages to existing pruning and quantization techniques? Or sparse training work[1]?

[1] Rigging the Lottery: Making All Tickets Winners.

**Questions:**

Check the Weaknesses.
More convincing experiments are needed.

---

> ### Author Response · Authors · 2023-11-20
>
> 1. I am a bit confused as to what you mean about only two models tested. We have tested both full-size and miniaturized VisionTransformers, a modern ResNet, the highly complex transformer model Oneformer, as well as the more traditional Autoformer for time series forecasting. We focused on transformers as they have become the de-facto standard for many use-cases and we show a variety of uses of them.  However, in the new comparison to other methods (see general comment), we present the un-tuned results for more a traditional ResNet-50 and VGG16.
> 2. That is correct. We would be happy to clarify that during revisions. This method is full- to low-rank method.
> 3. Please see the general comment for our response to this.

---

### Official Review · Reviewer_uXNh · 2023-10-31

**Soundness:** 2 fair
**Presentation:** 3 good
**Contribution:** 2 fair
**Rating:** 3
**Confidence:** 4

**Summary:**

In this paper, the authors propose a low-rank neural network update method (OIALR) via SVD decomposition.  Experiments on various network architectures and learning tasks show that the proposed OIALR achieves a slight accuracy loss with fewer parameters.

**Strengths:**

1. The paper is well-written and well-organized.

2. The authors provide experimental evaluations on various tasks and different network architectures.

**Weaknesses:**

1.  $\textbf{Marginal Contribution}$.

The orthogonal neural networks and low-rank neural networks are widely studied in the literature.  The technical contribution of this paper is marginal.  Low-rank neural networks and low-rank fine-tuning via SVD decomposition are not new.  The proposed Algorithm 2 seems to be an incremental variation compared with previous works.

2.  $\textbf{No discussion about the difference between the proposed algorithm and previous works}$

In this paper, the authors fail to provide a detailed discussion about the difference between the proposed algorithm and previous low-rank methods.  It is unclear what is the advantages and disadvantages of the proposed method compared with previous low-rank neural network methods.


3.  $\textbf{No comparison with related baselines}$

In this paper, there is no empirical comparison with related low-rank methods to support the advantage of the proposed method.  It is unconvincing to distinguish the proposed method from related low-rank methods without experimental compassion.

**Questions:**

Q1. Could the authors discuss the differences and advantages/disadvantages of the proposed method compared with related low-rank methods?

Q2.  Could the authors provide a comprehensive experimental comparison with low-rank neural network baselines?

Q3. What is the improvement of Algorithm 2 compared with a trivial baseline, i.e.,   low-rank approximation of a well-trained full-rank network?

Q4.  What is the size of the trainable parameter $\Sigma$ in Algorithm 2? It seems that the size of the $\Sigma$ is the same as the size of $W=U \Sigma V^\top $.  If so, what is the difference and advantage of Algorithm 2 compared with a standard full-rank training of $W$? In addition, what are the advantages/disadvantages of Algorithm 2 compared with full-rank training of $W$ and low-rank approximation/fine-tuning at the last step?

Q5.  In the paper, the authors argue the "Stability" of the proposed method. What is the formal definition of the "Stability"? Why does the proposed method achieve "Stability"  compared with other low-rank methods?

---

> ### Author Response · Authors · 2023-11-20
>
> Q3: As shown in related works, the trivial baseline does not perform well because the model depends upon the small singular values to perform well unless it is trained not to. See [1] for more details. This idea is common knowledge for low-rank (and full-rank to low-rank) training methods.
> Q4: The size of trainable parameters varies during training, as stated in Algorithm 2, as small singular values are removed, the size of the $\Sigma$ matrix is reduced.
> Q5:  In Section 3, we have expounded upon the metric 'Stability,' providing its conceptual underpinnings, visual representation in Figure 1, and a formal mathematical definition in Equation 1. It is imperative to note that, to the best of our knowledge, the manifestation and implications of this metric have not been reported previously. While the potential existence of stability in other methods is acknowledged, this exploration falls beyond the purview of our work and is contingent upon the nuances of each respective training method.
>
> - As for the advantages, disadvantages, and a comparison to other methods, we address this in the general comment.
>
> [1] T. N. Sainath, B. Kingsbury, V. Sindhwani, E. Arisoy and B. Ramabhadran, "Low-rank matrix factorization for Deep Neural Network training with high-dimensional output targets," _2013 IEEE International Conference on Acoustics, Speech and Signal Processing_, Vancouver, BC, Canada, 2013, pp. 6655-6659, doi: 10.1109/ICASSP.2013.6638949.

---

### Official Review · Reviewer_ViAF · 2023-11-01

**Soundness:** 3 good
**Presentation:** 3 good
**Contribution:** 3 good
**Rating:** 6
**Confidence:** 1

**Summary:**

This paper introduces an innovative training approach known as "Orthogonality-Informed Adaptive Low-Rank Neural Network Training." The method is rooted in the hypothesis that the orthogonal bases of the low-rank decomposition of neural network weights become more stable during training. By following standard tuning procedures, this proposed method outperforms conventional training setups. Additionally, the paper demonstrates the effectiveness of the tuned low-rank training procedure by applying it to enhance the performance of a state-of-the-art transformer model designed for time series prediction.

**Strengths:**

This paper showcases a commendable strength in its comprehensive and rigorous experimental methodology. The research rigorously evaluates various neural network architectures across diverse datasets, thus ensuring the generalizability and robustness of the proposed approaches. Notably, the experiments extend to the training of the OneFormer on the challenging CityScapes dataset, mini-ViT on CIFAR-10, and Autoformer on ETTm2, demonstrating the versatility and adaptability of the methods across distinct application domains and scenarios. This meticulous experimentation contributes significantly to the paper's credibility and the trustworthiness of its findings.

**Weaknesses:**

The progress towards achieving a state-of-the-art (SOTA) model is somewhat constrained. While the incremental improvements made in this work are commendable, a more extensive exploration of novel approaches or the inclusion of additional techniques may be necessary to achieve a substantial leap in performance that rivals the current SOTA models in the field.

**Questions:**

Why does the training time not exhibit a significant reduction even as the number of trainable parameters decreases substantially, from 100% to just 9.97%?

---

> ### Author Response · Authors · 2023-11-20
>
> - Pursuing SOTA models holds significance, yet it might be overly stringent to mandate that every scientific progress must distinctly contribute to enhancing SOTA models. The learning process during training remains a realm of ongoing investigation. This paper aims to provide some clarity on this and subsequently use that knowledge. In essence, revealing the intricacies of a model's training process and showing one way of how to use it, represents a step toward improving future models.
> - Regarding the training time: although there are less trainable parameters, there are still many calculations to do. the benefits of low-rank will not show themselves until the inner dimension is quite low. This is true for any low-rank method. As OIALR reduces the inner dimension over time, the benefits are not seen until late in training.

---

### Author Response · Authors · 2023-11-20
**General commnt to all reviewers**

Thank you all for taking the time for your reviews. We have taken the time to absorb them and run a few more experiments to hopefully shed more light. We will happily include these in the manuscript (as well as other minor changes) during revisions if accepted.

We have received a few reviews mentioning specifically the usage of OIALR against other low rank methods, while we are happy to provide this to the extent possible, the main focus of this work is section 3 where we show evidence that the orthogonal bases inherent to the multidimensional weights of a neural network stabilize to a single basis which that layer uses for the rest of training. This is a finding which, to our knowledge, has never been shown before. The objective of this work is *not* to outperform a full rank state of the art baseline.

The OIALR training method is different from other methods as it specifically maintains the orthogonal bases of the multi-dimensional weights during training. It shares some of its ideas with DLRT, as it trains only the $\Sigma$ matrix, but it does not train the other orthogonal bases directly. Furthermore, we are one of the only methods of this nature to train in less time than the baseline as most others require more training iterations.

Here are some results showing the differences in Top 1 accuracy to the baseline for various other low-rank, full- to low-rank, and pruning methods. The numbers shown here is the difference to the baseline model.
```
### Resnet 50 ImageNet-2012
OIALR      -1.54
DLRT[1]    -0.56
PP-2[2]    -0.8
PP-1[2]    -0.2
CP[3]      -1.4
SFP[4]     -0.2
ThiNet[5]  -1.5
RigL[6]    -2.2

### VGG6 ImageNet-2012
OIALR      -1.65
DLRT[1]    -2.19
PP-1[2]    -0.19
CP[3]      -1.8
ThiNet[5]  -0.47
RNP(3X)[7] -2.43

### VGG16 CIFAR10
OIALR      0.72
DLRT[1]    -1.89
GAL[8]     -1.87
LRNN[9]    -1.9
```

Training costs: The OIALR method is not written in optimized code. Most of the code for a NN in any language is written and optimized in low-level langauges, however the logic for OIALR is still written in Python. Furthermore, since the size of the memory buffers change during training, modern tools such as PyTorch's compile function cannot function. Since OIALR is already on par with (or reducing) training times when compared with optimized code, if it were to utilize low-level languages, it would further reduce the training time. Also, methods like RigL and DLRT both require many more forward-backward passes (x5 and x3 respectively) to achieve baseline accuracy.

[1] Steffen Schotthöfer, Emanuele Zangrando, Jonas Kusch, et al. Low-rank lottery tickets: finding efficient low-rank neural networks via matrix differential equations. Advances in Neural Information Processing Systems, 35:20051–20063, December 2022.

[2] P. Singh, V. Kumar Verma, P. Rai, and V. P. Namboodiri. Play and prune: Adaptive filter pruning for deep model compression. In Proceedings of the Twenty-Eighth International Joint Conference on Artificial Intelligence, IJCAI-19, pages 3460–3466. International Joint Conferences on Artificial Intelligence Organization, 7 2019.

[3] He, X. Zhang, and J. Sun. Channel pruning for accelerating very deep neural networks, 2017.

[4] He, G. Kang, X. Dong, Y. Fu, and Y. Yang. Soft filter pruning for accelerating deep convolutional neural networks, 2018.

[5] J.-H. Luo, J. Wu, and W. Lin. Thinet: A filter level pruning method for deep neural network compression, 2017

[6] Evci, Utku, et al. "Rigging the lottery: Making all tickets winners." _International Conference on Machine Learning_. PMLR, 2020.

[7] J. Lin, Y. Rao, J. Lu, and J. Zhou. Runtime neural pruning. In I. Guyon, U. V. Luxburg, S. Bengio, H. Wallach, R. Fergus, S. Vishwanathan, and R. Garnett, editors, Advances in Neural Information Processing Systems, volume 30. Curran Associates, Inc., 2017.

[8] S. Lin, R. Ji, C. Yan, B. Zhang, L. Cao, Q. Ye, F. Huang, and D. Doermann. Towards optimal structured CNN pruning via generative adversarial learning. In Proceedings of the IEEE/CVF Conference on Computer Vision and Pattern Recognition, pages 2790–2799, 2019

[9] Y. Idelbayev and M. A. Carreira-Perpiñán. Low-rank compression of neural nets: Learning the rank of each layer. In IEEE/CVF Conference on Computer Vision and Pattern Recognition (CVPR), pages 8046–8056, 2020.

---

### Meta-Review · Area_Chair_541L · 2023-12-14

**Metareview:**

This study presents a new low-rank training method of neural networks titled "Orthogonality-Informed Adaptive Low-Rank Neural Network Training." This technique is based on the premise that the orthogonal bases in the low-rank decomposition of neural network weights gain stability as training progresses.

Although the method has its own merits, most of the reviewers find that the experimental comparisons are not sufficient, and the improvements are marginal. We suggest the authors to incorporate this feedback for future submission

**Justification For Why Not Higher Score:**

The experimental results are not convincing enough.

**Justification For Why Not Lower Score:**

N/A

---

### Decision · Program_Chairs · 2024-01-16

Reject